# Mitochondria at the Heart of Sepsis: Mechanisms, Metabolism, and Sex Differences

**DOI:** 10.3390/ijms26094211

**Published:** 2025-04-29

**Authors:** John Q. Yap, Azadeh Nikouee, Jessie E. Lau, Gabriella Walsh, Qun Sophia Zang

**Affiliations:** 1Department of Surgery, Stritch School of Medicine, Loyola University Chicago, 2160 S. 1st Ave, Maywood, IL 60153, USA; jyap@luc.edu (J.Q.Y.); anikouee@luc.edu (A.N.); jlau8@luc.edu (J.E.L.); gwalsh2@luc.edu (G.W.); 2Burn & Shock Trauma Research Institute, Stritch School of Medicine, Loyola University Chicago, 2160 S. 1st Ave, Maywood, IL 60153, USA; 3Department of Microbiology and Immunology, Stritch School of Medicine, Loyola University Chicago, 2160 S. 1st Ave, Maywood, IL 60153, USA; 4Cardiovascular Research Institute, Stritch School of Medicine, Loyola University Chicago, 2160 S. 1st Ave, Maywood, IL 60153, USA

**Keywords:** sepsis, mitochondria, cardiac dysfunction

## Abstract

Sepsis is a life-threatening condition that occurs when the body is unable to effectively combat infection, leading to systemic inflammation and multi-organ failure. Interestingly, females exhibit lower sepsis incidence and improved clinical outcomes compared to males. However, the mechanisms underlying these sex-specific differences remain poorly understood. While sex hormones have been a primary focus, emerging evidence suggests that non-hormonal factors also play contributory roles. Despite sex differences in sepsis, clinical management is the same for both males and females, with treatment focused on combating infection using antibiotics and hemodynamic support through fluid therapy. However, even with these interventions, mortality remains high, highlighting the need for more effective and targeted therapeutic strategies. Sepsis-induced cardiomyopathy (SIC) is a key contributor to multi-organ failure and is characterized by left ventricular dilation and impaired cardiac contractility. In this review, we explore sex-specific differences in sepsis and SIC, with a particular focus on mitochondrial metabolism. Mitochondria generate the ATP required for cardiac function through fatty acid and glucose oxidation, and recent studies have revealed distinct metabolic profiles between males and females, which can further differ in the context of sepsis and SIC. Targeting these metabolic pathways could provide new avenues for sepsis treatment.

## 1. Introduction

Sepsis is a life-threatening condition that occurs when the body is unable to properly respond to infection, leading to systemic inflammation [1]. The early stage of sepsis is typically characterized by fever, elevated heart rate, increased respiratory rate, and minimal organ dysfunction [2]. In contrast, late-stage or severe sepsis involves a more exaggerated immune response, which can lead to vascular dysfunction and significant organ impairment [1,2]. In the United States, there are approximately 1.7 million cases of sepsis annually, with 350,000 deaths per year [3].

Sepsis can be community-acquired, arising from infections like pneumonia, urinary tract infections, and infected wounds, or hospital-acquired, resulting from medical procedures such as central line placement, ventilator use, catheter insertion, and surgery [4,5]. Pathogens that can trigger sepsis include bacteria, viruses, and fungi, but bacterial infections are by far the most common [6,7]. The most common bacteria that cause sepsis are *Escherichia coli* (*E. coli*), *Staphylococcus aureus* (*S. aureus*), *Klebsiella pneumoniae* (*K. pneumoniae*), *Streptococcus pneumoniae* (*S. pneumoniae*), and *Pseudomonas aeruginosa* (*P. aeruginosa*) [8,9]. Risk factors such as age, immunosuppression, comorbidities, and sex all contribute to predisposing individuals to sepsis [2,9]. Of particular interest are the sex-specific differences in sepsis, as males consistently exhibit a higher risk of developing sepsis and worse clinical outcomes [10,11,12,13,14].

## 2. Sex-Specific Outcomes During Sepsis

In 2003, a landmark study published in the *New England Journal of Medicine* analyzed 10,319,418 sepsis cases from 1979 to 2000 across multiple acute care hospitals in the United States [11]. The findings revealed that males were 28% more likely to develop sepsis compared to females [11]. More recently, researchers examined 343 million records globally across 204 countries in 2019 and found that males were more susceptible to infections from 11 of the top pathogens related to sepsis [9]. In addition to increased susceptibility to sepsis, both short-term and long-term mortality rates, as well as length of hospital stay, are significantly higher in men than in women [10,12,13,14].

Many have speculated that sex hormones play a role in the sex specific differences during sepsis. If estrogen and testosterone levels were the primary factors responsible for these differences, we would expect to see older women who have lower estrogen levels to experience susceptibility and mortality rates closer to those of older men. However, a large retrospective study involving 25,376 patients with bloodstream infections in Israel between 2018 and 2019 found that the difference in 30-day and one-year mortality rates between men and women was more pronounced in older patients compared to younger ones [10]. In contrast, a separate study at Beth Israel Deaconess Medical Center, which tracked 6134 adult sepsis patients in critical care units between 2001 and 2012, showed that the male–female mortality gap remained consistent across all age groups, with males consistently exhibiting higher mortality rates [12]. These contrasting findings highlight the need for further research into the sex-specific factors that influence susceptibility to sepsis and sepsis outcomes.

## 3. Pre-Clinical Models of Sepsis

Several pre-clinical models are commonly used to study sepsis, each with distinct advantages and limitations. The three most widely used models are lipopolysaccharide (LPS)-induced endotoxemia, cecal ligation and puncture (CLP), and pneumonia-induced sepsis [15]. These models allow researchers to investigate different aspects of sepsis pathophysiology.

In the LPS model, systemic inflammation is induced by administering LPS, a Gram-negative bacterial toxin [15,16,17,18]. LPS can be injected into mice to induce a systemic inflammatory response or used to stimulate immune cells in vitro. This model is widely used due to its simplicity and reproducibility [19]. Moreover, by adjusting LPS concentration and exposure time, researchers can examine different phases of sepsis [15]. However, because LPS exposure does not involve live bacteria, it does not fully recapitulate the complex host–pathogen interactions seen in sepsis [19]. Additionally, LPS-based models are primarily relevant to Gram-negative bacterial infections and do not account for Gram-positive or polymicrobial sepsis.

The CLP model is the most widely used in vivo model for polymicrobial sepsis and mimics intra-abdominal infections [18]. In this model, the cecum is ligated and punctured to allow leakage of intestinal contents into the peritoneal cavity, leading to sepsis [15,18]. CLP provides a more clinically relevant model of intra-abdominal sepsis, as it involves live bacterial infection and polymicrobial peritonitis [15,18]. The severity of sepsis can be modulated by adjusting factors such as the degree of cecal ligation, the number of punctures, and the size of the needle used [15]. While CLP better reflects the complexity of sepsis compared to LPS, it has relatively high variability in outcomes and mortality rates, which can complicate reproducibility across studies [19].

Pneumonia-induced sepsis models are used to study respiratory infections that cause sepsis [15]. This model is particularly relevant because pneumonia is one of the most common causes of sepsis in humans [15]. In this approach, live bacteria (such as *Streptococcus pneumoniae* or *Pseudomonas aeruginosa*) are introduced directly into the lungs, leading to localized infection that can progress to systemic sepsis [15]. The severity of sepsis can be controlled by varying the bacterial inoculum, making it possible to study different stages of disease progression [15]. However, this model typically involves a single pathogen species for infection, limiting its ability to capture the polymicrobial nature of sepsis [19].

A summary of the most commonly used pre-clinical models to study sepsis is presented in Table 1. Together, these models provide complementary insights into sepsis pathophysiology and reflect its most common causes, including respiratory infections (pneumonia model), intra-abdominal infections (CLP model), and bloodstream infections (LPS model). While each model has limitations, their combined use allows researchers to study the complex and heterogeneous nature of sepsis, ultimately aiding in the development of better therapeutic strategies.

## 4. Pathogenesis of Sepsis

The pathogenesis of sepsis involves two distinct but interconnected phases. The first is a hyperinflammatory phase, which is characterized by excessive immune cell activation and cytokine storm. This is followed by an immunosuppressive phase, in which immune cells become dysfunctional and or depleted, leading to impaired pathogen clearance and increased susceptibility to secondary infections. During the hyperinflammatory phase, pattern recognition receptors (PRRs) on immune cells recognize pathogen-associated molecular patterns (PAMPs), such as LPS on Gram-negative bacterial membranes, peptidoglycan from Gram-positive cell walls, bacterial DNA, and flagellin [20]. This triggers the release of pro-inflammatory cytokines like TNF-α, IL-1, and IL-6, which recruit additional immune cells to combat the infection. However, these pro-inflammatory cytokines can also cause tissue damage, triggering the release of damage-associated molecular patterns (DAMPs), such as ATP, mitochondrial DNA, and heat shock proteins. These endogenous molecules bind to pattern recognition receptors (PRRs) on immune cells, resulting in further immune cell recruitment and enhanced inflammation [21]. Excessive inflammation driven by extensive immune cell recruitment and a cytokine storm results in widespread endothelial damage, increased vascular permeability, dysregulated coagulation, and multi-organ failure [22]. Figure 1 provides a summary of the events that occur during both the hyperinflammatory and immunosuppressive phases.

Several innate immune cell subsets play critical roles in the host response to infection, including neutrophils, macrophages, dendritic cells, natural killer cells, and mast cells [23]. Among these, neutrophils and macrophages have been shown to play dominant roles in clearing bacterial pathogens [24]. Neutrophils, which are recruited by cytokines such as IL-8, IL-1, and TNF-α, contribute to bacterial clearance by releasing neutrophil extracellular traps to capture and kill pathogens [25]. Neutrophils also secrete pro-inflammatory cytokines and release chemokines to recruit additional immune cells, including macrophages [26]. Macrophages engulf and degrade pathogens through phagocytosis, present antigens to adaptive immune cells, and modulate inflammation by secreting both pro-inflammatory and anti-inflammatory cytokines [27]. Over the years, research has increasingly focused on macrophage polarization and functional plasticity. M1 macrophages are classically activated by IFN-γ and LPS and exhibit a pro-inflammatory phenotype [28,29]. They play a crucial role in host defense by producing pro-inflammatory cytokines, enhancing microbial killing through reactive oxygen and nitrogen species, and promoting Th1 immune responses [28,29]. In contrast, M2 macrophages are activated by IL-4 and IL-10 and contribute to tissue repair and homeostasis by clearing cellular debris, secreting anti-inflammatory cytokines, and promoting extracellular matrix remodeling and angiogenesis [28,29].

## 5. Sex Differences in Immune Regulation During Sepsis

Studies suggest that differences in immune responses between males and females contribute to sex-specific outcomes during sepsis [30,31,32,33]. One large-scale study found that patients with severe sepsis secondary to pneumonia had a significant increase in IL-6 and IL-10 levels one day after the onset of sepsis, with males showing higher levels compared to females [30]. However, from days 2 to 7, cytokine levels decreased and were similar in both males and females [30]. These findings suggest that sex may affect sepsis differently at different disease stages. Interestingly, significant sex differences were seen in levels of clotting factors, with females exhibiting higher levels of antithrombin III and factor IX [30]. This suggests that females may have a more tightly regulated coagulation response, potentially providing a protective advantage against sepsis-associated coagulopathy. In another study, ten young, healthy males and females were administered LPS, and immune responses were monitored over six hours [31]. The findings revealed that young females exhibited higher levels of inflammatory cytokines, including TNF-α and IL-6, compared to young males after LPS administration [31]. These findings suggest that sex hormones may contribute to sex differences in immune response during sepsis.

Pre-clinical studies have provided mechanistic insights into how sex hormones influence the immune response [32,33,34,35]. In a study from Ian Marriot’s group, ovariectomized (OVX) female mice exhibited reduced pro-inflammatory cytokine levels after LPS treatment compared to non-OVX female mice [32]. Additionally, flow cytometry analysis revealed that OVX females had lower toll-like receptor 4 (TLR4) expression on macrophages after LPS exposure, as well as reduced serum levels of LPS binding protein (LBP), a protein that facilitates LPS binding to its TLR4 receptor [32]. Conversely, OVX females treated with exogenous estrogen exhibited increased TLR4 expression and elevated levels of pro-inflammatory cytokines following LPS stimulation [32]. Furthermore, estrogen administration to OVX females worsened sepsis severity [32]. The same group showed in a separate study that TLR4 expression on macrophages was reduced after LPS exposure in gonadectomized male mice, and that when these mice were treated with testosterone, they exhibited an immunosuppressive phenotype [33].

Together, these studies highlight the contribution of sex hormones to immune differences in sepsis, with estrogen enhancing the immune response and testosterone exerting immunosuppressive effects. However, in older patients with diminished circulating sex hormones, additional factors likely contribute to sex differences in immune responses. These non-hormonal factors may include genetic, epigenetic, and metabolic influences, which remain largely unexplored. Notably, some studies have observed that estrogen levels rise comparably in males and females during sepsis, and that elevated estrogen levels are associated with increased mortality [36,37,38,39]. This suggests that other intrinsic biological factors may be at play. Further research is needed to fully elucidate these factors and their implications for personalized therapeutic approaches in sepsis management. Additionally, further investigation is needed to understand sex differences during the distinct hyperinflammatory and immunosuppressive phases of sepsis, as these phases may influence immune responses in a sex-dependent manner.

## 6. Metabolic Reprogramming in Immune Cells During Sepsis

Recently, studies have highlighted the role of immune cell metabolic reprogramming in sepsis [40,41,42,43]. In a recent study, Mihai Netea’s group comprehensively characterized these metabolic changes in different stages of sepsis [40]. To evaluate metabolic changes during the early stages of sepsis, the group utilized a human endotoxemia model, in which young healthy males received an intravenous dose of LPS [40]. Blood samples were collected from these patients before and four hours after LPS administration for gene expression analysis. As expected, LPS administration increased expression of genes associated with the acute inflammatory phase of sepsis, including pro-inflammatory cytokines and M1 macrophage markers [40]. LPS administration also increased the expression of genes linked to cytosolic glycolysis while suppressing those involved in mitochondrial oxidative phosphorylation and fatty acid oxidation [40]. These findings demonstrate that there is a switch from mitochondrial oxidative phosphorylation to cytosolic glycolysis during the early stages of sepsis (Figure 1).

Additionally, the authors analyzed gene expression in blood samples from healthy controls and patients with severe sepsis due to *E. coli* or Candida infections [40]. Patients with severe sepsis had increased expression of genes typically associated with M2 macrophages and reduced expression of those associated with M1 macrophages [40]. The decrease in M1 macrophages and increase in M2 macrophages is a hallmark of the immunosuppressive phase of sepsis [27,44]. Compared to the young subjects with early-stage sepsis, patients with severe sepsis had increased expression of genes related to mitochondrial oxidative phosphorylation, even with the upregulation of genes associated with mitochondrial dysfunction and downregulation of genes related to fatty acid oxidation [40]. These results highlight the metabolic changes that occur during different stages of sepsis (Figure 1). While this analysis offers valuable insights into metabolic reprogramming during sepsis, it is important to acknowledge that distinct cell types may exhibit varying metabolic responses in sepsis.

Growing interest has focused on whether metabolic reprogramming drives macrophage polarization during sepsis. In a key study, Luke O’Neill’s group analyzed metabolites in isolated bone marrow-derived macrophages (BMDMs) from young female mice, untreated or treated with LPS in vitro, and found that LPS significantly increased succinate levels in BMDMs [42]. Succinate inhibits the activity of prolyl hydroxylase domain-containing enzymes (PHD), leading to the stabilization and activation of the transcription factor HIF-1α [45]. This, in turn, promotes the expression of pro-inflammatory cytokines and the polarization of M1 macrophages [42,46]. In BMDMs treated with LPS and succinate, HIF1 and IL-1 levels increased significantly [42]. In a follow-up study, Luke O’Neill’s group showed that succinate signaling is dependent on reactive oxygen species (ROS) [41]. When ROS production was inhibited in vitro using MitoQ, a mitochondrial-specific ROS scavenger, there was a reduction in HIF-1α and IL-1 protein expression in BMDMs [41]. Furthermore, they found that inhibiting succinate dehydrogenase in vivo to decrease succinate levels also led to a reduction in IL-1 and HIF-1α expression [41]. These studies suggest that succinate accumulation during sepsis drives an M1 macrophage phenotype and emphasize the role of metabolic intermediates, like succinate, in regulating macrophage polarization.

During sepsis, studies have shown that the transition to immunosuppression drives metabolic reprogramming in macrophages, favoring M2 polarization [27,44]. Recent studies suggest epigenetic mechanisms influence metabolic reprogramming in immune cells [47,48,49]. One study found that IL-4 treatment in BMDMs promotes SUMO-specific protease 1 (SENP1) translocation to the mitochondria, leading to decreased SUMOylation of Sirtuin 3 (SIRT3), a mitochondrial deacetylase [48]. Reduced SUMOylation enhanced SIRT3-dependent deacetylation of glutamate dehydrogenase 1 (GLUD1) and increased its enzymatic activity [48]. GLUD1 catalyzes the conversion of glutamate to α-ketoglutarate (αKG), a key metabolic intermediate that influences macrophage polarization [48]. Treatment of BMDMs with IL-4 increased αKG entry into the nucleus, resulting in increased histone demethylation at H3K27me3 and M2 polarization, shown by increased M2 macrophages on flow cytometry and upregulation of M2-specific genes [48]. While αKG can also be converted to succinate, this study found IL-4 treatment resulted in a low αKG-to-succinate ratio, suggesting that αKG is primarily directed toward epigenetic regulation rather than metabolism at this stage [48]. This aligns with previous findings from Luke O’Neill’s group, which demonstrated that succinate accumulation supports an M1 phenotype [41,42]. The study also found that IL-4 stimulation increased mitochondrial oxidative phosphorylation in BMDMs and that inhibiting fatty acid oxidation (FAO) with etomoxir did not affect M2 polarization through the SENP1-SIRT3 pathway, suggesting this pathway functions independently of FAO [48].

Studies have also shown that pyruvate dehydrogenase (PDH) may play an important role in immune cell polarization [43,50]. PDH is a mitochondrial enzyme that converts pyruvate produced from cytosolic glycolysis into acetyl coenzyme A (acetyl-CoA). Acetyl-CoA enters the TCA cycle, where it plays an important role in mitochondrial oxidative phosphorylation [51]. During sepsis, immune cells upregulate the expression of pyruvate dehydrogenase kinase 1 (PDK1), a kinase that phosphorylates and inhibits PDH [52,53]. In a recent study, flow cytometry analysis revealed that treatment with the PDK inhibitor dichloroacetate (DCA) in mice subjected to CLP increased the ratio of anti-inflammatory to pro-inflammatory immune cells [43]. This suggests that enhancing PDC activity and mitochondrial oxidative phosphorylation drives an anti-inflammatory immune phenotype.

Overall, these findings emphasize how metabolic reprogramming drives immune cell phenotype throughout the stages of sepsis. This metabolic reprogramming alters the levels of key intermediates, such as αKG and succinate, which regulate gene expression. Continued investigation is needed to better understand the mechanisms by which metabolic reprogramming drives immune cell phenotypes. Additionally, while recent research has primarily focused on macrophages, further studies are needed to explore how these metabolic changes vary across different immune cell types.

## 7. Coagulation Dysregulation in Sepsis

In addition to combating infection, immune cells play a crucial role in initiating coagulation during sepsis. The coagulation cascade consists of the intrinsic and extrinsic pathways, both of which converge on a common pathway that activates Factor X (FX). Activated FX (FXa) then catalyzes the conversion of prothrombin to thrombin. Thrombin cleaves fibrinogen to fibrin, which promotes clot formation at the site of infection, in conjunction with activated platelets and neutrophils [54,55]. The extrinsic pathway is initiated by the release of tissue factor (TF) from neutrophils, macrophages, and endothelial cells upon binding PAMPs. This interaction activates nuclear factor kappa B (NF-κB) signaling, leading to an upregulation of TF expression [54,56,57,58]. TF is then released by pyroptosis, a highly inflammatory form of programmed cell death. In this process, caspases cleave gasdermin D (GSDMD), which inserts into the cell membrane to form pores, facilitating TF release and enhanced TF activity [59]. In contrast, the intrinsic pathway is activated when coagulation factor XII (FXII) binds to products of endothelial injury, such as DNA, RNA, and components of atherosclerotic plaques [54,55].

During the early stages of sepsis, laboratory markers often show elevated procoagulant proteins, such as tissue factor (TF), and decreased antithrombotic proteins, including antithrombin III (ATIII) [60,61,62]. Additionally, platelet counts decrease due to increased incorporation into clots [60,63]. Initially, clot formation serves an adaptive role in preventing excessive bleeding, but if dysregulated, it can lead to sepsis-induced coagulopathy (SIC). A hallmark of SIC is impaired fibrinolysis, driven by increased release of plasminogen activator inhibitor-1 (PAI-1) from endothelial cells [60]. PAI-1 expression is upregulated in response to pro-inflammatory cytokines such as TNF-α, leading to fibrinolytic shutdown [60,64,65]. If SIC persists, it can progress to overt disseminated intravascular coagulation (DIC), which is characterized by severe depletion of fibrinogen, platelets, and anticoagulant proteins [54,60]. The clinical phenotype of DIC can be thrombotic, fibrinolytic, or mixed. Thrombotic DIC is marked by excessive clot formation with suppressed fibrinolysis, whereas fibrinolytic DIC is associated with excessive fibrin breakdown, leading to bleeding and increased plasma fibrinolytic markers such as D-dimer [54,60]. Severe thrombocytopenia significantly increases mortality risk. Studies suggest that up to 30% of patients with septic shock may develop overt DIC within three days of admission [66], underscoring the need for early identification and intervention to prevent hemostatic failure and associated mortality.

## 8. Endothelial Dysfunction During Sepsis

During sepsis, endothelial dysfunction plays a central role in the development of organ dysfunction. Normally, the endothelium maintains vascular tone, barrier integrity, and anticoagulant properties, largely through its protective glycocalyx layer. However, in the setting of systemic inflammation, this barrier becomes rapidly disrupted, contributing to capillary leak, leukocyte adhesion, thrombosis, and tissue hypoperfusion. The glycocalyx layer is primarily composed of heparan sulfate proteoglycans (HSPs) [67]. The predominant HSPs that make up the glycocalyx are syndecans, which play a crucial role in binding anticoagulation factors like antithrombin III and antioxidants such as superoxide dismutase (SOD) [67,68]. As such, glycocalyx integrity is essential for maintaining vascular barrier function, preventing oxidative stress, and promoting anticoagulation. Retrospective studies have shown elevated serum syndecan protein levels in patients during sepsis, indicating glycocalyx degradation [69,70,71,72,73]. Furthermore, increased serum syndecan levels were associated with higher mortality in septic patients [71,72,73], highlighting their clinical relevance. One study using human adipose microvascular endothelial cells demonstrated that LPS-induced oxidative stress enhances histone deacetylase (HDAC) activity, which suppresses the expression of tissue inhibitors of metalloproteinases (TIMPs) [74]. TIMPs typically inhibit matrix metalloproteinases (MMPs), but when their activity is reduced, MMPs cleave syndecan proteins, resulting in glycocalyx degradation [74]. Once the glycocalyx is degraded, endothelial cells are exposed to inflammatory mediators, leading to endothelial cell apoptosis. Studies using various endothelial cell lines have shown that exposure to LPS or TNFα increased cytokine production, oxidative stress, and apoptosis by activating mitogen-activated protein kinase (MAPK) and p38 signaling pathways [75,76,77,78].

Pre-clinical studies have demonstrated that activation of MAPK and p38 signaling pathways during acute inflammation enhances the expression of nitric oxide synthase (NOS) in endothelial cells, leading to increased production of nitric oxide (NO) [79,80,81]. NO is a potent vasodilator, and clinical studies using NOS inhibitors, such as N(G)-nitro-L-arginine methyl ester (L-NAME) and hemoglobin polyoxyethylene (PHP), have shown transient improvements in blood pressure [82,83,84]. However, prolonged use of these inhibitors has been associated with increased mortality due to cardiac complications [83,84]. As a result, more recent research has focused on strategies to enhance NO bioavailability. For example, nanoparticle-mediated NO delivery has been shown to improve vascular permeability and modulate inflammation in septic mice [85]. Additionally, an emerging therapeutic target is asymmetric dimethylarginine (ADMA), an endogenous NOS inhibitor that is elevated in patients with sepsis and associated with increased mortality [86,87,88,89]. ADMA is degraded by dimethylarginine dimethylaminohydrolase (DDAH), which enhances NO signaling [86]. Overexpression of DDAH has shown potential in improving endothelial function in human lung microvascular endothelial cells treated with LPS [90].

Clinical studies have also investigated the effects of increasing NO production during sepsis, with a particular focus on nitroglycerin, an NO-releasing drug commonly used to treat angina [91,92]. In 2002, a small interventional study was published in *The Lancet*, which involved eight patients with septic shock [91], and orthogonal polarization spectral (OPS) imaging was used to assess blood flow in sublingual micro-vessels. Patients received a 0.5 mg intravenous loading dose of nitroglycerin, and images of the sublingual micro-vessels were captured before and 2 min after administration. Imaging revealed increased microvascular circulation after the nitroglycerin administration [91]. Nitroglycerin infusion was then continued at 0.5–4 mg/h, after which clinical signs were improved, and seven patients were discharged alive [91].

More recently, a phase 3 trial of nitroglycerin in sepsis was conducted, which included 35 patients with severe sepsis and 35 control patients [92]. In this study, 2 mg of NO was administered within 30 min, followed by a continuous infusion of 2 mg/h for 23.5 h [92]. OPS imaging of sublingual micro-vessels was performed before administration and at 30 min, 2 h, 12 h, and 24 h after administration [92]. Interestingly, there were no differences in microvascular circulation between the placebo and NO groups at any time point [92]. Moreover, the study found that NO administration was associated with an increased 60-day mortality rate [92]. The differences between these studies may be due to several factors, including patient populations (severe sepsis vs. septic shock), dosing regimens, and the timing of imaging. These investigations into modulating vascular tone highlight the uncertainty of whether systemic vasodilation or vasoconstriction is more beneficial during sepsis.

In addition to the effects of inflammation on endothelial cell apoptosis and NO production, studies in various endothelial cell lines have shown that inflammation disrupts both cell-to-cell and cell-to-tissue junctions. Bannerman et al. demonstrated that LPS or TNFα exposure in bovine pulmonary artery endothelial cells activates caspases, leading to cleavage of β-catenin and focal adhesion proteins, endothelial detachment, and cell death [93,94]. Caspase inhibition with Z-Val-Ala-Asp (OMe) Fluoromethylketone (Z-VAD) reduced these effects, preserving cell adhesion and viability [94]. Similarly, Blum et al. used immunofluorescence microscopy to show that TNF-α and IFN-γ exposure in microvascular endothelial cells caused actin filament reorganization, disrupting occludins and tight junctions, exacerbating vascular leak and hypotension [95]. While these studies provide valuable insights into endothelial dysfunction during sepsis, further research utilizing in vivo models and system-specific endothelial cells would offer a more comprehensive understanding of the specific mechanisms underlying endothelial disruption during sepsis.

## 9. Management of Sepsis

Current treatments for sepsis focus on managing hemodynamic instability and the underlying infection [2,22]. These include the use of vasopressors to counteract hypotension and antibiotics to address the infection [2,22]. Current treatment protocols do not differentiate between males and females, despite emerging evidence that sex-specific differences may influence the pathophysiology and response to treatment in sepsis. This is a significant gap, as accumulating evidence suggests that sex-specific variations in immune response, disease progression, and treatment efficacy may impact clinical outcomes.

In the clinical setting, commonly used biomarkers to assess sepsis include C-reactive protein (CRP), procalcitonin (PCT), and serum lactate. However, these single biomarkers have demonstrated only modest prognostic value. Their specificity is often inadequate, leading to false positives in non-septic inflammatory conditions, while sensitivity can be variable. Moreover, most of these markers have not been validated for use in the very early phases of sepsis, limiting their usefulness in early triage and decision-making [96]. Beyond biomarkers, scoring systems such as Systemic Inflammatory Response Syndrome (SIRS) and Sequential Organ Failure Assessment (SOFA) are widely used to identify and track sepsis severity. SIRS is highly sensitive but lacks specificity, often identifying patients with non-infectious inflammation. In contrast, SOFA correlates more closely with organ dysfunction and outcomes but may miss early or less overt cases [96,97]. Together, these tools provide a framework for sepsis assessment, yet neither captures the heterogeneity of host immune responses.

Given these limitations, efforts have intensified to develop novel approaches for patient stratification [98,99,100]. In a recent multi-center study, 266 emergency department (ED) patients across four continents were enrolled within two hours of ED admission based on suspected sepsis, defined by the presence of at least two SIRS criteria and clinical suspicion of infection. Whole-blood RNA sequencing followed by machine learning analysis enabled the identification of five distinct endotypes, each characterized by approximately 200 unique genes, specific biological pathways, and differing clinical outcomes [99].

Notably, two of these endotypes, called neutrophilic suppressive (NPS) and inflammatory (INF), were associated with increased severity and mortality. The NPS endotype showed heightened expression of genes involved in neutrophil degranulation, increased glycolytic activity, and a suppressed IFN-γ response. Conversely, the INF group demonstrated elevated expression of inflammatory genes and TLR4 signaling [99]. These mechanistic distinctions suggest that patients in the NPS group may benefit from immunostimulatory therapies, such as IFN-γ administration or neutrophil-targeting agents, while INF patients might respond better to anti-inflammatory strategies.

Collectively, these findings underscore the potential of transcriptomic profiling to complement existing tools such as lactate, CRP, SOFA, and SIRS in refining patient triage and guiding personalized treatment strategies. Ultimately, future research should prioritize precision approaches that account for patient-specific variables, including sex, immune phenotype, and comorbidities, to improve sepsis outcomes.

## 10. Sepsis-Induced Cardiomyopathy (SIC)

Sepsis-induced cardiomyopathy (SIC) is a key component of multi-organ dysfunction syndrome (MODS) and significantly contributes to hemodynamic instability, impaired fluid balance, and inadequate oxygen delivery [101]. These factors exacerbate organ dysfunction and increase mortality [101]. While SIC lacks a formal definition, it is typically characterized by left ventricular dilation, reduced cardiac output, and impaired myocardial contractility [101]. Many of the risk factors for SIC overlap with those of sepsis, including age, comorbidities such as diabetes, and sex [101,102,103].

In line with data on clinical outcomes of mortality, females generally show better cardiac outcomes during sepsis than males [102,103,104], experiencing fewer ventricular arrhythmias [102], fewer occurrences of atrial fibrillation [104], and better cardiac function based on echocardiograms [103]. However, clinical studies comparing sex differences in sepsis-induced cardiomyopathy are limited, and the results are often criticized due to the variability of cardiac function during sepsis. As a result, mortality rates are often used as more reliable outcome measures.

## 11. Pathogenesis of SIC

### 11.1. Sepsis-Induced Mitochondrial Dysfunction in Cardiomyocytes

Cardiomyocytes are the primary cells responsible for cardiac contractility [105]. In the healthy heart, contractions are initiated by an electrical signal, resulting in activation of L-type calcium channels, allowing calcium entry into the cell [105]. Increased cytosolic calcium is sensed by ryanodine receptors on the sarcoplasmic reticulum (SR), triggering the release of additional calcium from the SR [105]. Calcium binds to troponin, enabling actin and myosin to form cross-bridges and contract [105]. This process relies on ATP, which is produced by mitochondria, highlighting the importance of mitochondrial health and function [105]. During sepsis, these processes are disrupted through several mechanisms, resulting in impaired cardiac contractility.

Several pre-clinical studies on sepsis have demonstrated that sepsis activates MAPK and nuclear factor kappa-light-chain-enhancer of activated B cells (NFκB) pathways in the heart, leading to increased production of pro-inflammatory cytokines [106,107,108,109]. Increased inflammation impairs mitochondrial function, disrupting ATP production and hindering the heart’s ability to meet its energy demands [106,107,110,111]. Along with functional disturbances, inflammation induces structural changes in mitochondria, including alterations to cristae morphology and increased mitochondrial fission, as observed by electron microscopy [106,107,112,113,114] and confirmed by elevated expression of fission proteins such as DRP1 and reduced expression of fusion proteins like MFN2 [106,107]. Furthermore, inflammation diminishes mitochondrial biogenesis, as evidenced by fewer mitochondria on electron microscopy [115] and reduced expression of biogenesis-related genes such as Peroxisome Proliferator-Activated Receptor Gamma Coactivator 1-Alpha (PGC-1α) [115,116].

During SIC, dysfunctional mitochondria release cytochrome c, triggering caspase-mediated apoptosis [110,117,118,119,120,121]. A pre-clinical study found that losartan, an angiotensin II receptor blocker (ARB), reduced pro-inflammatory and pro-apoptotic signaling in LPS-treated mice by inhibiting MAPK and NFκB pathways [107]. Furthermore, a large retrospective study in Taiwan, involving 52,982 septic patients, demonstrated improved clinical outcomes and decreased 30- and 90-day mortality in patients receiving ARB therapy [122]. Interestingly, males showed a greater reduction in 30-day mortality compared to females in the subgroup analysis [122]. This could potentially be attributed to greater resolution of inflammation, as suggested by the pre-clinical study on the mechanisms of ARBs.

A recent study by Xin et al. identified miR-101-3p as a regulator of dual-specificity phosphatase 1 (DUSP1), a MAPK phosphatase in SIC [108]. Upregulation of this miRNA in septic patients correlated with worsened cardiac function, suggesting that targeting miR-101-3p could restore DUSP1 activity, suppress MAPK signaling, and mitigate SIC [108]. Indeed, mice treated with miR-101-3p inhibitors exhibited decreased MAPK signaling, reduced inflammatory cytokine production, and improved cardiac outcomes during LPS-induced SIC [108]. Interest in miRNAs as therapeutic targets is growing, and studies like this highlight their potential for future SIC treatments.

Pre-clinical studies have shown that damaged mitochondria produce increased levels of reactive oxygen species (ROS) during SIC [109,113,123,124,125]. This leads to oxidation of critical calcium-handling proteins, including ryanodine receptors [123] and SERCA [126], resulting in calcium leakage, overload, and reduced contractility [123,124,126]. Calcium overload further impairs mitochondrial function by inducing permeability transition pore (mPTP) opening, leading to mitochondrial membrane depolarization and cardiomyocyte death [127]. Inhibition of ROS production, either through NOX2 inhibition [124] or the administration of antioxidants such as high-dose vitamin D [128] or the mitochondria-specific ROS scavenger MitoQ [125], has been shown to restore calcium handling and improve cardiac function in pre-clinical models of sepsis [124,125,128]. Given the damaging effects of ROS on cardiac tissue and the promising results from pre-clinical studies, antioxidants have garnered significant interest as potential therapeutic agents for improving sepsis outcomes. A small retrospective study of 94 patients with severe sepsis found that those receiving high-dose vitamin D, in combination with hydrocortisone and thiamine, had reduced mortality compared to those not receiving vitamin D [129]. Furthermore, two small clinical trials [130,131], one phase 1 trial involving 24 patients with severe sepsis [131] and another involving 28 patients with septic shock [130], showed that high-dose intravenous vitamin D administration significantly decreased 28-day mortality. These initial studies highlight the potential of ROS scavenging as a therapeutic approach. Additionally, exploring outcome differences between broad-spectrum ROS scavengers and mitochondrial-specific scavengers would be valuable.

Regarding sex specific differences in SIC pathogenesis, pre-clinical studies have shown that estrogen provides cardiac protection during SIC by enhancing resistance to oxidative stress [132,133] and by reducing activation of key inflammatory pathways, including NFκB and MAPK signaling [132,133,134]. Furthermore, in a trauma–hemorrhage rat model of SIC, estrogen activated the phosphoinositide 3 kinase (PI3K) and protein kinase B (AKT) pathway, which reduced apoptosis [135]. Estrogen also reduced monocyte adhesion and infiltration into the heart in pre-clinical models of SIC, thereby decreasing cytokine production and mitigating inflammation [132,136]. These actions by estrogen collectively contribute to improved cardiac outcomes in pre-clinical models of SIC, supporting its protective role in females. However, significant gaps remain, as most sepsis research has focused on male subjects, leaving the molecular mechanisms underlying sex differences largely unexplored. Additionally, the role of non-hormonal factors in these disparities remains poorly defined.

### 11.2. Sepsis-Induced Alterations in Myocardial Metabolism

Clinical studies have shown that sepsis disrupts cardiac metabolism by decreasing fatty acid and glucose oxidation and increasing cytosolic glycolysis [137,138]. Because the heart primarily depends on fatty acid and glucose oxidation for energy production [139], these alterations in cardiac metabolism compromise ATP generation and cardiac function. Additionally, the accumulation of metabolic intermediates in the heart during sepsis can further damage cardiac tissue [140]. In the healthy heart, fatty acids are transported into cardiomyocytes by transport proteins such as CD36, fatty acid-binding protein (FABPpm), and fatty acid transport protein (FATP), while glucose uptake occurs primarily through glucose transporter 1 (GLUT1) and glucose transporter 4 (GLUT4) [139]. Once inside cardiomyocytes, fatty acids are converted to acyl coenzyme A (acyl-CoA) by acyl-CoA synthetase and transported by carnitine palmitoyl transferase I (CPT1) into the mitochondria for β-oxidation, generating acetyl-CoA [139]. Acetyl-CoA enters the tricarboxylic acid (TCA) cycle in the mitochondrial matrix, producing high-energy molecules such as NADH and FADH2, which fuel the electron transport chain to drive ATP production [51]. Glucose, on the other hand, first undergoes glycolysis in the cytoplasm to form pyruvate, which is then converted to acetyl-CoA by the PDH and fed into the TCA cycle, where it follows the same metabolic fate as acetyl-CoA derived from fatty acids [51]. The ATP produced through these pathways is critical for maintaining cardiac function by ensuring proper excitation–contraction coupling, ion homeostasis, and contractile force generation [141]. Furthermore, the heart maintains metabolic flexibility, allowing it to adjust substrate utilization based on physiological demands [142].

Under normal physiological conditions, the heart primarily derives about 60% of its energy from fatty acid oxidation and 40% from glucose metabolism [142,143]. These percentages stem from experiments by Dr. Garry Lopaschuk and colleagues in the late 1900s, which used carbon tracing in male rats to assess myocardial substrate utilization [143]. However, the experiments were only conducted in male animals, overlooking potential sex-based differences in cardiac metabolism. It was not until 2007 that Dr. Linda Peterson and colleagues explored sex-based differences in human cardiac metabolism using positron emission tomography (PET) imaging [144,145]. These studies revealed that females exhibited lower baseline myocardial glucose utilization and higher fatty acid metabolism compared to males [144,145], demonstrating that cardiac metabolism is more complex and dynamic than previously appreciated.

Given that fatty acid oxidation serves as the primary energy source for the heart, early research predominantly focused on its role in SIC. Pre-clinical studies of SIC demonstrated that inflammation decreases fatty acid oxidation in the heart and downregulates genes involved in fatty acid metabolism and transport [115,146,147,148]. Notably, interventions that enhance fatty acid metabolism, such as c-Jun N-terminal kinase (JNK) inhibition or treatment with rosiglitazone, a PPAR-γ agonist, were shown to improve cardiac function [115,146]. Additionally, overexpression of peroxisome proliferator-activated receptor gamma coactivator 1 beta (PGC1β), a key regulator of fatty acid metabolism and mitochondrial biogenesis, restored mitochondrial function and fatty acid metabolism in LPS-treated mice [148]. Interestingly, these interventions improved cardiac function without significantly altering inflammation [115,146,148], suggesting that metabolic dysfunction in SIC occurs downstream of inflammatory pathways such as MAPK and p38 signaling.

Despite initial focus on fatty acid metabolism, more recent studies have shifted to glucose metabolism, which is impaired during sepsis due to decreased glucose transport and oxidation [137,138,146,149]. Because glucose requires less energy to metabolize than fatty acids [150], enhancing glucose utilization could help reduce the overall energy demand during metabolic stress. Multiple studies using different models of SIC have consistently shown upregulation of myocardial pyruvate dehydrogenase kinase 4 (PDK4) [115,146,149,151,152], an important regulator of glucose metabolism. PDK4 is a mitochondrial matrix protein that decreases glucose oxidation by phosphorylating and inhibiting PDH [153]. In the human body, there are four PDK isoforms that phosphorylate and inhibit PDH [153]. These isoforms are expressed in a tissue-dependent manner [153], and PDK4 has been shown to be highly expressed in the heart [153].

Research by Dr. Luke Swzeda and colleagues demonstrated that a high-fat diet in mice elevated PDK4 expression, leading to increased PDH phosphorylation and decreased PDH activity, thereby promoting reliance on fatty acid metabolism [154]. Restoring mice to a normal diet reduced PDK4 expression [154]. Additionally, Dr. Szweda and colleagues demonstrated that when glucose metabolism is active, PDK4 detaches from PDH, allowing for its degradation by the Lon protease [155]. Other studies have shown that PDK4 expression is regulated by forkhead box O (FOXO) transcription factors, particularly in the context of high-fat diet (HFD)-induced cardiac dysfunction [156,157,158]. FOXO activation in HFD mice has been demonstrated to increase PDK4 expression, while FOXO knockout enhanced glucose oxidation and improved cardiac function in a diabetic cardiomyopathy model [158]. Additionally, a separate study identified an AKT-p38-FOXO signaling pathway, where activation of AKT and p38 in a diabetic cardiomyopathy model promoted FOXO nuclear translocation, leading to the upregulation of PDK4 [156].

More recently, the role of PDK4 in SIC has been further elucidated. In a study by Shimada et al., mice with cecal ligation puncture (CLP)-induced sepsis showed significantly increased myocardial PDK4 expression at both the RNA and protein levels, leading to increased PDH phosphorylation and decreased PDH activity [149]. Mitochondria isolated from cardiac tissue in CLP-treated mice exhibited reduced function, as evidenced by lower oxygen consumption rates (OCRs) in Seahorse assays [149]. Tagtian et al. extended these findings, showing that in H9C2 cells derived from embryonic rat heart tissue, LPS treatment increased PDK4 expression, reduced mitochondrial membrane potential, decreased OCR, promoted mitochondrial fragmentation, and increased lactate production [152]. In a follow-up study, the same group used a CLP mouse model to demonstrate that PDK4 knockdown by adeno-associated virus (AAV)-mediated gene silencing or pharmacological inhibition of PDK4 with dichloroacetate (DCA) restored pyruvate metabolism, improved mitochondrial function, and preserved cardiac performance [151]. Interestingly, recent evidence suggests that PDK4 contributes to mitochondrial dysfunction beyond its role in PDH inhibition. One study found that PDK4 can independently drive mitochondrial fragmentation in human embryonic kidney (HEK) cells [159], while another demonstrated that PDK4 inhibits autophagy in breast cancer cells [160]. These studies suggest that targeting PDK4 to improve PDH activity, along with mitochondrial structure and function, may provide a promising therapeutic approach for SIC. A summary of the key metabolic changes that occur during sepsis is shown in Figure 2.

### 11.3. Sepsis-Induced Autophagy Is Not Sufficient to Rescue Cardiac Function

When organelles such as the mitochondria become damaged, the cell initiates a protective mechanism called autophagy, which directs damaged organelles for degradation to prevent further cellular damage [161]. When this process specifically targets damaged mitochondria, it is referred to as mitophagy. In this process, damaged mitochondria increase the expression of mitophagy-related proteins such as PTEN-induced kinase 1 (PINK1) on the outer membrane [162]. PINK subsequently recruits and activates the ubiquitin ligase Parkin by phosphorylation [162]. Parkin ubiquitinates mitophagy-related proteins expressed on the mitochondrial membrane [162]. These ubiquitinated proteins interact with p62 and LC3-II, which play key roles in the formation of autophagosomes [162]. Autophagosomes then fuse with lysosomes, where their contents are degraded [161]. Notably, increased LC3-II levels suggest the formation of autophagosomes and mitophagosomes in the early stage of autophagy, while a reduction in p62 levels indicates the subsequent degradation process via lysosomes, reflecting the later phase of autophagic flux [163,164]. During sepsis, the heart enhances autophagy and mitophagy to help clear away dysfunctional mitochondria [112,114].

Recently, our group published a study demonstrating that autophagy levels depend on the severity of infection [112]. Autophagy, as assessed by LC3-II and p62 levels in cardiac tissue from mice, increased in a dose-dependent manner with LPS treatment, peaking at 2.5 mg/kg [112]. However, at higher doses of 5, 10, and 15 mg/kg, autophagy decreased but was still higher compared to healthy control mice [112]. These results suggest that at moderate levels of infection, the autophagic process is upregulated, helping to clear damaged organelles and mitigate cellular stress. However, when the infection severity exceeds a certain threshold, the autophagic response becomes overwhelmed and disrupted, likely due to the excessive damage that surpasses the capacity of the autophagic machinery.

We further demonstrated that cardiac specific overexpression of Beclin-1, a key autophagy protein, enhanced autophagy, rescued cardiac function, reduced inflammation, and restored mitochondrial integrity in mice treated with LPS [112]. These findings suggest that although autophagy is activated during SIC, the level of activation remains insufficient to fully restore cardiac function. The mechanisms of autophagy induction and its capacity are still being investigated. A transcriptomic study utilizing cardiac tissue from sepsis patients and control patients revealed that sepsis induces upregulation of genes that both inhibit and promote autophagy [165]. Upregulation of inhibitory genes may prevent the heart from achieving optimal autophagy levels necessary to restore cardiac function. As such, there has been growing interest in identifying targets to enhance cardiac autophagy, such as Beclin-1. Another promising target for enhancing cardiac autophagy is Transmembrane BAX Inhibitor Motif Containing 1 (TMBIM1), a stress-related protein. In a pre-clinical study, LPS treatment in rats increased TMBIM1 expression in the heart [166]. TMBIM1 was also shown to promote mitophagy in neonatal rat cardiomyocytes (NRCM) by facilitating the recruitment of Parkin to damaged mitochondria [166]. Overexpression of TMBIM1 in rat hearts enhanced mitophagy, reduced apoptosis, alleviated cardiac inflammation, and improved cardiac function [166].

In addition to exploring potential targets for enhancing autophagy, there has been growing interest in currently available medications that could potentially promote autophagy. Among the medications being studied is Levosimendan, an inotropic agent that is currently used to treat acute exacerbations of severe chronic heart failure [167,168]. A recent pre-clinical study demonstrated that Levosimendan treatment increased cardiac mitophagy in LPS-treated mice, resulting in improved mitochondrial health, reduced cardiac inflammation, and enhanced cardiac function [169]. Similarly, clinical studies have reported that Levosimendan improves cardiac function and reduces inflammation in patients with sepsis [170]. In addition to its effects potential effects on mitophagy, Levosimendan has been shown to protect the heart during ischemia-reperfusion injury by activating ATP-sensitive potassium (K_ATP) channels in both the heart and smooth muscle, which helps to maintain mitochondrial membrane potential, support mitochondrial function, and reduce oxidative stress [168].

### 11.4. Sepsis-Induced Cardiac Fibrosis

During sepsis, inflammation triggers myocardial fibrosis, a key factor in the development of both systolic and diastolic dysfunction. In mouse models, treatment with LPS resulted in increased cardiac fibrosis as demonstrated by Hematoxylin and Eosin staining (H&E staining) [171] and upregulated the expression of fibrosis-related genes, including MMP-2, MMP-9, collagen, and TIMPs [171,172]. A recent large-scale drug screening study by Zhan et al. identified the LPS co-receptor Myeloid Differentiation Factor 2 (MD2) as a therapeutic target for sepsis-induced fibrosis [173]. The study highlighted Artesunate, a drug used to treat malaria, as the most effective compound in inducing iPSC-derived cardiac fibroblast apoptosis without affecting cardiomyocyte or endothelial cell viability [173]. Molecular simulations revealed that Artesunate disrupts MD2 and TLR4 interactions, thereby inhibiting extracellular signal-regulated kinase (ERK)-mediated fibrotic gene activation [173]. Because cardiac fibrosis usually develops later in the progression of SIC, early intervention could offer therapeutic benefits.

## 12. Summary

Currently, treatments for sepsis are largely limited to fluid resuscitation and antibiotics [2], but the high mortality rate due to sepsis-induced complications such as MODS suggests a pressing need for better approaches. Major contributors to MODS include hemodynamic instability, impaired fluid imbalances, and inadequate oxygen delivery, which are exacerbated by SIC [101]. Therefore, improving cardiac outcomes during sepsis could lead to better outcomes.

A simplified model of how sepsis disrupts cardiac function is summarized in Figure 3, where infection activates pro-inflammatory signaling cascades such as the MAPK pathway, resulting in production of pro-inflammatory cytokines, which amplify inflammation and disrupt mitochondrial function, leading to increased ROS, impaired myocardial metabolism, decreased energy production, increased release of DAMPS, heightened inflammation, and impaired cardiac function. While targeting upstream inflammatory pathways presents one therapeutic approach, it is essential to consider the balance between immune cell recruitment and mitigating the harmful consequences of excessive inflammation, such as mitochondrial dysfunction. Given the pivotal role of mitochondria in cardiac function, improving cardiac mitochondrial function and myocardial metabolism during SIC may prove to be a promising target for therapeutic intervention. Additionally, several recent studies have explored enhancing autophagy as a therapeutic strategy, yielding promising pre-clinical results. Table 2 summarizes the potential therapeutic targets discussed in this review. Continued research into the mechanisms of mitochondrial dysfunction, along with the development of targeted therapies, could lead to more effective treatments for sepsis patients. Additionally, as the world moves toward personalized medicine, understanding how different patients respond to sepsis is crucial. One key factor is the notable difference in how males and females respond to infection. A summary of the male and female differences discussed here is provided in Table 3. A deeper understanding of the mechanisms underlying these sex differences will be vital as individualized treatments become more prevalent.

## Figures and Tables

**Figure 1 ijms-26-04211-f001:**
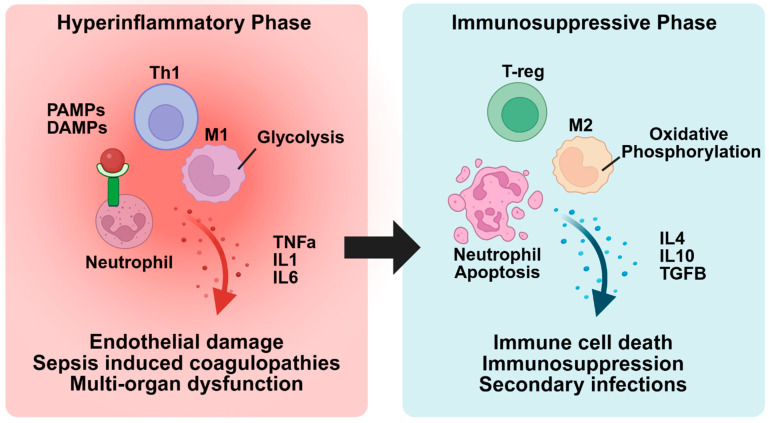
Transition from hyperinflammation to immunosuppression in sepsis. In the hyperinflammatory phase, PAMPs and DAMPs bind to receptors on immune cells, leading to their activation and a heightened inflammatory response. This can result in endothelial cell damage, coagulopathies, and multi-organ dysfunction. As sepsis progresses, the immune system transitions to the immunosuppressive phase, characterized by inflammatory immune cell death, upregulation of anti-inflammatory immune cells, and a suppressed immune response. This phase increases the risk of secondary infections and further exacerbates the risk of morbidity and mortality.

**Figure 2 ijms-26-04211-f002:**
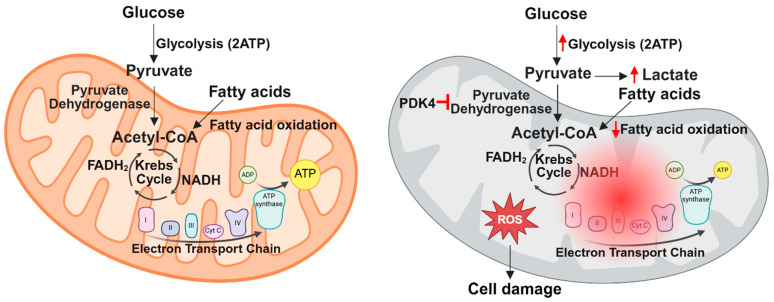
Summary of metabolic changes in the heart during sepsis. Under normal conditions, the heart primarily generates ATP through fatty acid and glucose metabolism within the mitochondria. However, sepsis disrupts mitochondrial function, leading to decreased fatty acid and glucose oxidation and increased cytosolic glycolysis due to PDK4 stimulation. These defects in cardiac metabolism cause increased ROS production and cell damage.

**Figure 3 ijms-26-04211-f003:**
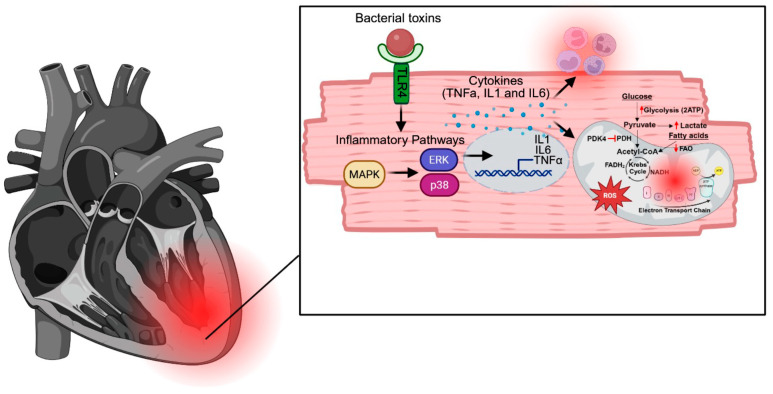
A simplified model of how sepsis disrupts cardiac function. Infection activates pro-inflammatory signaling cascades, including the MAPK pathway, leading to the production of pro-inflammatory cytokines. These cytokines amplify inflammation and disrupt mitochondrial function, resulting in increased ROS and impaired myocardial metabolism. The subsequent decrease in energy production and increased release of DAMPs further exacerbate inflammation, ultimately impairing cardiac function.

**Table 1 ijms-26-04211-t001:** Summary of common pre-clinical sepsis models. This table provides an overview of the three most commonly used pre-clinical models of sepsis: LPS-induced endotoxemia, CLP, and pneumonia-induced sepsis. It summarizes their descriptions, underlying pathophysiology, advantages, and limitations.

Model	Lipopolysaccharide (LPS)	Cecal Ligation Puncture (CLP)	Pneumonia
Description	Purified LPS from gram-negative bacteria is administered, resulting in systemic inflammation and sepsis like symptoms.	The cecum is ligated and punctured, releasing intestinal contents into the peritoneum, resulting in polymicrobial sepsis.	Bacteria such as *S. pneumonia* or *K. pneumoniae* are inoculated into the lungs to induce sepsis.
Pathophysiology	LPS binds to TLR4 on immune cells and non-immune cells across organ systems, triggering cytokine release, leading to local and systemic inflammation.	CLP causes peritonitits, leading to systemic inflammatory response syndrome, multi organ dysfunction and immune response dysregulation.	Pnemonia can lead to acute inflammation, alveolar damage and systemic infection, causing sepsis and multi-organ failure.
Advantages	This model is simple and reproducible.	This model is clinically representative of intra-abdominal infections.	This model is clinically representative of pneumonia induced sepsis.
Disadvantages	This model does not represent a true infection, as it involves only a toxin from Gram-negative bacteria and therefore cannot capture the full complexity of sepsis.	There is high variability in outcomes and mortality rates.	This model generally uses only one pathogen strain for infection.

**Table 2 ijms-26-04211-t002:** Summary of potential therapeutic targets and agents for SIC. This table highlights key mechanisms involved in sepsis-induced cardiac dysfunction, including endothelial degradation, inflammatory signaling, oxidative stress, myocardial metabolism, autophagy, and fibrosis. Potential therapeutic targets within each pathway are listed alongside therapeutic agents that have been assessed, with corresponding reported outcomes. NA refers to “Not assessed in this review” but potentially evaluated in other research.

Pathogenic Mechanism	Potential Therapeutic Targets	Therapeutic Agents Reviewed	Clinical Outcomes of Therapeutic Agents Reviewed	References
Endothelial Degradation	TIMPs, MMPs, ADMA, DDHA, NOS, Caspases	L-NAME, PHP, and Nitroglycerin: increase vascular dilation	Mixed results on blood circulation and patient outcomes	[86,87,88,95,96]
Inflammatory Signaling	MAPK, p38, NFκB, miR-101-3p, TLR4, Caspases	ARBs: decreases MAPK and NFκB signaling in the heart.	Decreased inflammation and mortality	[126]
Oxidative Stress	NOX, ROS	High dose Vitamin D: ROS scavenger	Decreased mortality and organ dysfunction	[133,134,135]
Metabolic Dysregulation	PGC1β, JNK, PDK4, PDH, GLUT1, GLUT4	NA	NA	NA
Autophagy	Beclin-1, TMBIM1, LC3, p62, PINK, Parkin	Levosimendan: increases autophagy in the heart	Improved cardiac function	[173]
Fibrosis	MD2, MMPs, TIMPs, Collagen, ERK	Artusenate: inhibits MD2, decreasing fibrosis	NA	NA

**Table 3 ijms-26-04211-t003:** Summary of sex differences during sepsis. This table highlights the key differences between males and females in response to sepsis, including variations in incidence, mortality rates, immune response, and cardiac metabolism.

Category	Males	Females
Sepsis Incidence	Higher incidence	Lower incidence
Mortality Rate	Higher mortality	Lower mortality
Early Immune Response	Elevated levels of pro-inflammatory cytokines (e.g., TNF-α, IL-6)	Moderated pro-inflammatory response
Cardiac Substrate Preference	Preferential glucose utilization	Preferential fatty acid oxidation
Mitochondrial Function	More susceptible to sepsis-induced mitochondrial dysfunction	Relative preservation of mitochondrial function
Therapeutic Considerations	May benefit from strategies targeting excessive inflammation and glycolysis	May benefit from supporting fatty acid oxidation and mitochondrial integrity

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
