# Peer review of "Mitochondria at the Heart of Sepsis: Mechanisms, Metabolism, and Sex Differences"

_ijms, 2025, doi:10.3390/ijms26094211_

Round 1

Reviewer 1 Report

Comments and Suggestions for Authors

This is a timely an important review that is a well constructed and an includes considerable depth of concept. However, I conceive limitations in the following areas:

First, I was impatient before getting to the subjects  of sex differences and mitochondrial biology and biochemistry. Thus, I suggest that the opening five pages may  cause reader thoughts to  stray and could be deleted.  I recommend the narrative begin on page 6 or at least at the area in which sex  differences are discussed and the review of mitochondrial biology and biochemistry begins. 

Second, many of the references were published before 2020 and there is need for updating to the now.   For example the  subject of immunometabolic  paralysis during a sepsis resistance to tolerance energy repression is not covered as it relates to nutrient selection of glucose oxidation vs fatty acid oxidation and possibly branched chain amino acids dehydrogenase  for TCA cycle electron and hydrogen shuttling to respiration complex IV and V heart energetics. Research publications by the Mihai Netea group in Radboud should be included and Stacpoole et al. associated progress updated to sepsis. How do mitochondria retrieve homeostasis in seps survivors. 

Third, I suggest redoing the figure and perhaps using more than one since the narrative might seem  dense for some readers. Within the context of the figures, I suggest that including a mitochondria anatomic scheme and molecular biology might help, to include nutrient TCA cycle entry and the proton motive  force into complex IV and V. I like, focusing on the heart, but perhaps systemic alterations and bar energetics should be included to those associated with weight loss and thermal dysregulation.

Fourth, a table of unanswered questions and knowledge gaps might improve narrative content and stimulate reader depth of knowledge. 

Author Response

Reviewer 1:

  1. First, I was impatient before getting to the subjects of sex differences and mitochondrial biology and biochemistry. Thus, I suggest that the opening five pages may cause reader thoughts to stray and could be deleted. I recommend the narrative begin on page 6 or at least at the area in which sex differences are discussed and the review of mitochondrial biology and biochemistry begins.

We appreciate the reviewer’s suggestion and understand the concern regarding the pacing of the manuscript. We included the introductory discussion on sepsis and bacterial infections to provide necessary context, as bacterial infections remain the most common cause of sepsis. However, we recognize the importance of focusing on the core themes of sex differences and mitochondrial biology. The discussion on sex differences begins on page 5, shortly after the introduction. In response to the reviewer’s feedback, we have further strengthened these sections by adding a dedicated discussion in the “Pathogenesis of Sepsis” section titled "Sex Differences in Immune Regulation During Sepsis" (pages 9–11), as well as a new section on mitochondrial biology, "Metabolic Reprogramming in Immune Cells During Sepsis" (pages 11–14). We hope these additions address the reviewer’s concerns and enhance the focus and clarity of the manuscript.

  1. Second, many of the references were published before 2020 and there is a need for updating to the now. For example, the subject of immunometabolic paralysis during a sepsis resistance to tolerance energy repression is not covered as it relates to the nutrient selection of glucose oxidation vs fatty acid oxidation and possibly branched-chain amino acids dehydrogenase for TCA cycle electron and hydrogen shuttling to respiration complex IV and V heart energetics. Research publications by the Mihai Netea group in Radboud should be included and Stacpoole et al. associated progress updated to sepsis. How do mitochondria retrieve homeostasis in sepsis survivors.

We thank the reviewer for this valuable suggestion. In response, we have expanded our discussion of the immunosuppressive phase of sepsis in the Pathogenesis of Sepsis section (pages 7-8) and have included a new section on mitochondrial biology “Metabolic re-programing in immune cells during sepsis” which highlights recent insights into immunometabolic paralysis and the transition from the hyperinflammatory phase to the immunosuppressive phase (pages 11-14). We now discuss nutrient selection and specifically the roles of glycolysis versus oxidative phosphorylation in the context of immune cell energetics during sepsis. We have incorporated relevant studies from the Mihai Netea group (Radboud University), Stacpoole et al. and Luke O'Neill. Additionally, we have cited several new publications throughout the manuscript to ensure that the review reflects current advancements in the field.

  1. Third, I suggest redoing the figure and perhaps using more than one since the narrative might seem dense for some readers. Within the context of the figures, I suggest that including a mitochondria anatomic scheme and molecular biology might help, to include nutrient TCA cycle entry and the proton motive force into complex IV and V. I like, focusing on the heart, but perhaps systemic alterations and bar energetics should be included to those associated with weight loss and thermal dysregulation.

We thank the reviewer for this helpful suggestion. In response, we have added a new dedicated figure (Figure 2) illustrating the metabolic changes that occur in the heart during sepsis. This figure includes mitochondrial anatomy and highlights impaired oxidative phosphorylation and fatty acid oxidation, increased cytosolic glycolysis and increased production of reactive oxygen species during sepsis. We believe this addition will help readers better visualize and understand the metabolic alterations in the septic heart.

  1. Fourth, a table of unanswered questions and knowledge gaps might improve narrative content and stimulate the reader depth of knowledge.

We thank the reviewer for this insightful suggestion. In response, we have emphasized unanswered questions and knowledge gaps throughout the manuscript by italicizing them to draw the reader’s attention. Additionally, we have added two new tables: one summarizing pre-clinical models for studying sepsis and another highlighting sex differences in sepsis discussed in this review.

Reviewer 2

 1. Limited Discussion on Pathophysiology: The section on the pathophysiology of sepsis is relatively brief and lacks a thorough discussion of key underlying mechanisms. Expanding this section to include more details on the immune response, endothelial dysfunction, and cellular metabolism would significantly strengthen the manuscript. A deeper analysis of how mitochondrial dysfunction interplays with inflammatory pathways would be beneficial.

We thank the reviewer for this valuable suggestion. In response, we have expanded the Pathogenesis of Sepsis section (pages 7–8) to provide a more comprehensive overview of the immunosuppressive phase. Additionally, we have added a new section titled “Metabolic Reprogramming in Immune Cells During Sepsis” (pages 11–14), which discusses recent insights into immunometabolic paralysis and the metabolic shifts from glycolysis to oxidative phosphorylation during the transition from hyperinflammation to immunosuppression. We have also included two new dedicated sections: “Endothelial Dysfunction During Sepsis” and “Coagulation Dysregulation in Sepsis,” which delve into the vascular and coagulative abnormalities observed in septic patients. We hope these additions enhance the depth of the manuscript and contribute to a more thorough and engaging discussion of sepsis pathophysiology.

  1. Lack of a Comprehensive Figure on Sex Differences: While the manuscript discusses sex differences in sepsis outcomes and mitochondrial function, the only figure presented does not effectively illustrate these differences. A new figure that visually summarizes the proposed sex-specific mechanisms would enhance clarity. For example, a side-by-side comparison of mitochondrial metabolism in male versus female patients, highlighting hormonal influences and metabolic adaptations, would be a valuable addition.

We thank the reviewer for this insightful suggestion. In response, we have added a table (Table 3) highlighting sex differences in sepsis discussed in this review.

  1. Insufficient Number of Figures and Tables: Given the complexity of the topic, a single figure and one table do not adequately support the depth of information presented. Additional graphical summaries, such as:
    • A schematic representation of mitochondrial dysfunction in sepsis
    • A flowchart of key metabolic pathways disrupted during sepsis
    • A summary table of sex-specific findings in mitochondrial metabolism would help to clarify key concepts and improve readability.

//We thank the reviewer for your suggestions. In response, we have created a new figure (Figure 1) that illustrates key events in the transition from the hyperinflammatory phase to the immunosuppressive phase of sepsis. Additionally, we have added a dedicated figure (Figure 2) depicting the metabolic changes that occur in the heart during sepsis. This figure includes mitochondrial anatomy and highlights impaired oxidative phosphorylation and fatty acid oxidation, increased cytosolic glycolysis, and elevated production of reactive oxygen species during sepsis. We believe these visual additions will enhance reader’s understanding of both immune and metabolic alterations during sepsis. Furthermore, we have included two new tables: one summarizing commonly used pre-clinical models for studying sepsis, and another outlining the sex differences in sepsis discussed throughout the review.

  1. Limited Discussion on Therapeutic Implications: The manuscript highlights several therapeutic targets but does not discuss their clinical applicability in sufficient detail. Expanding the discussion to include:
    • Current clinical trials exploring metabolic interventions in sepsis
    • Potential sex-specific therapeutic strategies
    • Challenges in translating mitochondrial-targeted therapies to clinical practice would provide a more complete perspective on the topic.

We thank the reviewer for your suggestions. At present, there are no clinical trials specifically targeting metabolic interventions in sepsis. However, in Table 3, we highlight the potential role of metabolism in driving sex-specific outcomes. For example, males may benefit from therapies aimed at reducing excessive inflammation and glycolysis, while females may benefit from strategies that support fatty acid oxidation and mitochondrial integrity. Additionally, in the Management of Sepsis section (pages 19–21), we have expanded our discussion on patient heterogeneity as a major challenge in sepsis treatment and introduced the concept of endotyping as a potential strategy to better stratify and manage patients.

  1. Unclear Relationship Between Therapeutic Agents and Targets: The effects of the therapeutic agents listed in the table are not clearly explained, and in some cases, they do not directly correspond to the therapeutic targets mentioned. A more detailed explanation of how each agent acts on its respective target, along with supporting references, would improve the clarity and usefulness of this section.

We thank the reviewer for your helpful suggestion. In response, we have added the proposed mechanisms of action for each therapeutic agent as reported in the studies reviewed. Additionally, we have included the corresponding references to support each entry. We hope these additions improve the clarity and utility of the table for readers.

Reviewer 2 Report

Comments and Suggestions for Authors

Overall the study is an interesting reviw of an important topic. Hoever I have some concers that should be adressed. 

  1. Limited Discussion on Pathophysiology
    The section on the pathophysiology of sepsis is relatively brief and lacks a thorough discussion of key underlying mechanisms. Expanding this section to include more details on the immune response, endothelial dysfunction, and cellular metabolism would significantly strengthen the manuscript. A deeper analysis of how mitochondrial dysfunction interplays with inflammatory pathways would be beneficial.
  2. Lack of a Comprehensive Figure on Sex Differences
    While the manuscript discusses sex differences in sepsis outcomes and mitochondrial function, the only figure presented does not effectively illustrate these differences. A new figure that visually summarizes the proposed sex-specific mechanisms would enhance clarity. For example, a side-by-side comparison of mitochondrial metabolism in male versus female patients, highlighting hormonal influences and metabolic adaptations, would be a valuable addition.
  3. Insufficient Number of Figures and Tables
    Given the complexity of the topic, a single figure and one table do not adequately support the depth of information presented. Additional graphical summaries, such as:
    • A schematic representation of mitochondrial dysfunction in sepsis
    • A flowchart of key metabolic pathways disrupted during sepsis
    • A summary table of sex-specific findings in mitochondrial metabolism would help to clarify key concepts and improve readability.
  4. Limited Discussion on Therapeutic Implications
    The manuscript highlights several therapeutic targets but does not discuss their clinical applicability in sufficient detail. Expanding the discussion to include:
    • Current clinical trials exploring metabolic interventions in sepsis
    • Potential sex-specific therapeutic strategies
    • Challenges in translating mitochondrial-targeted therapies to clinical practice would provide a more complete perspective on the topic.
  5. Unclear Relationship Between Therapeutic Agents and Targets
    The effects of the therapeutic agents listed in the table are not clearly explained, and in some cases, they do not directly correspond to the therapeutic targets mentioned. A more detailed explanation of how each agent acts on its respective target, along with supporting references, would improve the clarity and usefulness of this section.

Author Response

Reviewer 2

  1. Limited Discussion on Pathophysiology: The section on the pathophysiology of sepsis is relatively brief and lacks a thorough discussion of key underlying mechanisms. Expanding this section to include more details on the immune response, endothelial dysfunction, and cellular metabolism would significantly strengthen the manuscript. A deeper analysis of how mitochondrial dysfunction interplays with inflammatory pathways would be beneficial.

We thank the reviewer for this valuable suggestion. In response, we have expanded the Pathogenesis of Sepsis section (pages 7–8) to provide a more comprehensive overview of the immunosuppressive phase. Additionally, we have added a new section titled “Metabolic Reprogramming in Immune Cells During Sepsis” (pages 11–14), which discusses recent insights into immunometabolic paralysis and the metabolic shifts from glycolysis to oxidative phosphorylation during the transition from hyperinflammation to immunosuppression. We have also included two new dedicated sections: “Endothelial Dysfunction During Sepsis” and “Coagulation Dysregulation in Sepsis,” which delve into the vascular and coagulative abnormalities observed in septic patients. We hope these additions enhance the depth of the manuscript and contribute to a more thorough and engaging discussion of sepsis pathophysiology.

  1. Lack of a Comprehensive Figure on Sex Differences: While the manuscript discusses sex differences in sepsis outcomes and mitochondrial function, the only figure presented does not effectively illustrate these differences. A new figure that visually summarizes the proposed sex-specific mechanisms would enhance clarity. For example, a side-by-side comparison of mitochondrial metabolism in male versus female patients, highlighting hormonal influences and metabolic adaptations, would be a valuable addition.

We thank the reviewer for this insightful suggestion. In response, we have added a table (Table 3) highlighting sex differences in sepsis discussed in this review.

  1. Insufficient Number of Figures and Tables: Given the complexity of the topic, a single figure and one table do not adequately support the depth of information presented. Additional graphical summaries, such as:
    • A schematic representation of mitochondrial dysfunction in sepsis
    • A flowchart of key metabolic pathways disrupted during sepsis
    • A summary table of sex-specific findings in mitochondrial metabolism would help to clarify key concepts and improve readability.

//We thank the reviewer for your suggestions. In response, we have created a new figure (Figure 1) that illustrates key events in the transition from the hyperinflammatory phase to the immunosuppressive phase of sepsis. Additionally, we have added a dedicated figure (Figure 2) depicting the metabolic changes that occur in the heart during sepsis. This figure includes mitochondrial anatomy and highlights impaired oxidative phosphorylation and fatty acid oxidation, increased cytosolic glycolysis, and elevated production of reactive oxygen species during sepsis. We believe these visual additions will enhance reader’s understanding of both immune and metabolic alterations during sepsis. Furthermore, we have included two new tables: one summarizing commonly used pre-clinical models for studying sepsis, and another outlining the sex differences in sepsis discussed throughout the review.

  1. Limited Discussion on Therapeutic Implications: The manuscript highlights several therapeutic targets but does not discuss their clinical applicability in sufficient detail. Expanding the discussion to include:
    • Current clinical trials exploring metabolic interventions in sepsis
    • Potential sex-specific therapeutic strategies
    • Challenges in translating mitochondrial-targeted therapies to clinical practice would provide a more complete perspective on the topic.

We thank the reviewer for your suggestions. At present, there are no clinical trials specifically targeting metabolic interventions in sepsis. However, in Table 3, we highlight the potential role of metabolism in driving sex-specific outcomes. For example, males may benefit from therapies aimed at reducing excessive inflammation and glycolysis, while females may benefit from strategies that support fatty acid oxidation and mitochondrial integrity. Additionally, in the Management of Sepsis section (pages 19–21), we have expanded our discussion on patient heterogeneity as a major challenge in sepsis treatment and introduced the concept of endotyping as a potential strategy to better stratify and manage patients.

  1. Unclear Relationship Between Therapeutic Agents and Targets: The effects of the therapeutic agents listed in the table are not clearly explained, and in some cases, they do not directly correspond to the therapeutic targets mentioned. A more detailed explanation of how each agent acts on its respective target, along with supporting references, would improve the clarity and usefulness of this section.

We thank the reviewer for your helpful suggestion. In response, we have added the proposed mechanisms of action for each therapeutic agent as reported in the studies reviewed. Additionally, we have included the corresponding references to support each entry. We hope these additions improve the clarity and utility of the table for readers.

Round 2

Reviewer 1 Report

Comments and Suggestions for Authors

The alteration is much improved or would be at quite long if not too long. Nevertheless, the subject of mitochondria and how it might be separated in male and female, as well as what’s going on in systemic, septic shock and sepsis syndromes is quite important and needs further discussion and investigation. The authors are to be applauded for such a hard job long work and I’ll be at I would prefer a shortening to target mitochondria rather than an immuno metabolism, that is a personal opinion that may not be shared by the editor and the second reviewer

Author Response

We sincerely thank the reviewer for the kind words. In response to Reviewer 1’s recommendation to shorten the manuscript, we have removed the section discussing shifting microbial trends in sepsis to sharpen the focus on mitochondria, metabolism, and sex differences. We retained the section on immunometabolism based on the reviewer’s earlier suggestion. In this section, we have clarified that oxidative phosphorylation is a mitochondrial process while glycolysis is a cytosolic process, thereby strengthening the link between changes in immunometabolism and mitochondrial function. We hope these changes adequately address the reviewer’s concerns.
